# Fine-tuning of BERT Model to Accurately Predict Drug–Target Interactions

**DOI:** 10.3390/pharmaceutics14081710

**Published:** 2022-08-16

**Authors:** Hyeunseok Kang, Sungwoo Goo, Hyunjung Lee, Jung-woo Chae, Hwi-yeol Yun, Sangkeun Jung

**Affiliations:** 1Department of Bio-AI Convergence, Chungnam National University, Daejeon 34134, Korea; 2College of Pharmacy, Chungnam National University, Daejeon 34134, Korea; 3Department of Computer Convergence, Chungnam National University, Daejeon 34134, Korea

**Keywords:** drug–target interaction, bidirectional encoder representations from transformers (BERT), ChemBERTa, ProBert, pretrained model, self-supervised learning

## Abstract

The identification of optimal drug candidates is very important in drug discovery. Researchers in biology and computational sciences have sought to use machine learning (ML) to efficiently predict drug–target interactions (DTIs). In recent years, according to the emerging usefulness of pretrained models in natural language process (NLPs), pretrained models are being developed for chemical compounds and target proteins. This study sought to improve DTI predictive models using a Bidirectional Encoder Representations from the Transformers (BERT)-pretrained model, ChemBERTa, for chemical compounds. Pretraining features the use of a simplified molecular-input line-entry system (SMILES). We also employ the pretrained ProBERT for target proteins (pretraining employed the amino acid sequences). The BIOSNAP, DAVIS, and BindingDB databases (DBs) were used (alone or together) for learning. The final model, taught by both ChemBERTa and ProtBert and the integrated DBs, afforded the best DTI predictive performance to date based on the receiver operating characteristic area under the curve (AUC) and precision-recall-AUC values compared with previous models. The performance of the final model was verified using a specific case study on 13 pairs of subtrates and the metabolic enzyme cytochrome P450 (CYP). The final model afforded excellent DTI prediction. As the real-world interactions between drugs and target proteins are expected to exhibit specific patterns, pretraining with ChemBERTa and ProtBert could teach such patterns. Learning the patterns of such interactions would enhance DTI accuracy if learning employs large, well-balanced datasets that cover all relationships between drugs and target proteins.

## 1. Introduction

Drug discovery is typically expensive, labor-intensive, and inefficient. To overcome these limitations, there is interest in methods that could increase the efficiency of finding compounds that may biochemically interact with specific targets. As knowledge on drugs, their targets, and their interactions accumulate, various computational methods have been developed to predict possible drug–target interactions (DTIs) to aid in drug discovery. In particular, machine learning (ML) methods have demonstrated an encouraging performance in DTI prediction, which used to take drug and protein data as inputs; the drugs are small molecules with a low molecular weight (≤1000 daltons) and may regulate a biological process. Proteins are mostly receptors that receive and transduce signals that may be integrated into biological systems. The model that receives the input data approaches the DTIs as a classification problem and makes predictions using deep learning models, such as Deep Neural Networks (DNNs) [1], Deep Belief Networks (DBNs) [2], Convolutional Neural Networks (CNNs) [3,4,5], and transformer networks [6].

Recently, pretrained language models were found to be optimal for the completion of many natural language processing (NLP) tasks [7]; Transformer-based models have performed impressively in many applications. Inspired by these successes, various transformer-based representation learning approaches have been introduced into the fields of chemical analysis and drug discovery. ChemBERTa [8] and ProtBert [9] are representative pretrained language models for chemical compounds and proteins, respectively; both were constructed using Bidirectional Encoder Representations from Transformers (BERT). ChemBERTa uses simplified molecular-input line-entry system (SMILES)-formatted data to learn chemical structural information, whereas ProtBert attempts to learn protein representations based on amino acid sequences.

In addition to the optimal choice of a pretrained model for DTI prediction, the dataset used to train the ML model requires attention. Basically, datasets that analyze protein–protein, drug–protein, or drug–drug interactions all have unique characteristics due to differences in the data. For instance, although BIOSNAP, DAVIS, and Binding DB have been widely used, BIOSNAP is a network database that uses a graph structure to describe not only the affinity between drugs and targets but also the activation and inhibition relationships between them; DAVIS is focused on kinases and their ligands. The interaction between 72 kinase inhibitors and 442 kinases covers >80% of the human catalytic protein kinome; binding BD is dedicated to drug–target interaction. The latter DB thus considers not only binding but also Ki, IC50, and Kd values. As model performance depends on the training dataset, an optimal dataset must be chosen for a DTI prediction model.

As no previous effort has been made to verify the utility of a pretrained model in terms of DTI prediction, or the dataset dependency, the main aims of this study were the development of a DTI prediction model pretrained with ChemBERTa and ProtBERT, which analyzes information on chemical compounds and proteins, respectively, and the effective merging of the datasets to ensure accurate DTI prediction; we then compare our model to previous models.

## 2. Datasets and Methods

### 2.1. Dataset Configuration

A number of machine-learning-based DTI prediction methods have been proposed [10]. Various DTI datasets have also been introduced for training and testing, including Drug Bank [11], KEGG [12], BIOSNAP [13], DAVIS [14], BindingDB [15], and others. Here, we used the BIOSNAP, DAVIS, and BindingDB datasets after careful consideration of their characteristics. BIOSNAP is a drug–target interaction dataset that contains information about the genes (i.e., those encoding proteins) that are targeted by drugs that are available on the US market, and DAVIS is a dataset focusing on kinases and their inhibitors, which are important proteins in intracellular signaling. Finally, BindingDB is an open dataset that mainly focuses on the interactions between proteins considered candidate drug targets and small druglike ligands and has a wider scope than the other two datasets. BIOSNAP, DAVIS, and BindingDB were used as datasets for the training and evaluation of our method. The BIOSNAP dataset consists of 4510 drug nodes and 2181 protein nodes. From DrugBank, 13,741 DTI pairs were used. Similarly, DAVIS includes information on 68 drugs and 379 proteins, and BindingDB includes 10,665 drugs and 1413 proteins. DAVIS and BindingDB regard an interaction as positive when the $K_{d}$ value of the drug and protein is less than 30 units. The dataset is structured as shown in Table 1.

### 2.2. Model Configuration

Pretraining language models have been confirmed to be effective in many natural language processing tasks. Among the various pretrained models used in natural language processing, the transformer [6] model learns context and meaning by tracking relationships within sequential data, such as words in a sentence. In addition to natural language, this is used to learn the relationships between tokens in the data that are expressed as text combinations. For example, a model can learn chemical information in the form of sentences (SMILES is one such model).

Among the transformer models, BERT [16] has performed impressively in a wide range of applications. Transformer is a model with an encoder–decoder structure, in which attention is used but not recurrence or convolution. It consists of multiple encoder and decoder layers; one encoder layer is composed of two lower layers (self-attention and feed forward layers), and the self-attention layer learns token dependency in the input sequence with the scaled dot-product attention of multihead attention. The feedforward layer is used to transmit the vector output of all tokens that were obtained from the self-attention layer to the next encoder layer as a vector input of the same position.

BERT has a model structure in which multiple encoder layers are stacked. It is pretrained using a Masked Language Model (MLM) and Next Sentence Prediction (NSP). MLM involves the random masking of several tokens in the input sequence and prediction of the masked token from the contextual information provided by the surrounding tokens. General language models predict masked tokens by analyzing previous tokens, while BERT predicts masked tokens by considering all tokens. Therefore, token dependencies are learned from the surrounding tokens. NSP determines whether two input sentences are continuous and learns the dependency between them.

In BERT or BERT-related transformer settings, the vector from a specially prepared token—[CLS]—contains the information of the input sequence obtained from pretraining; [CLS] is a special classification token, which is added in front of every input sequence to express the aggregated sequence information in the classification and regression problem [16]. As with natural language, transformer-based feature embedding and processing techniques were recently introduced for chemical compounds and amino acid sequences [17]. ChemBERTa [8] is based on a RoBERTa [18] model pretrained with large-scale chemical compound data expressed in the SMILES format. ProtBert [9] was developed by pretraining a BERT model using an amino acid sequence dataset to capture useful protein features in vector form.

Here, we used two transformer-based encoders for chemical compounds and amino acid sequences to solve the DTI prediction problem (Figure 1). One transformer encodes a SMILES input sequence to represent chemical compound information, and the other captures useful protein-related information in vector form from amino acid sequences. The sequencewise information of chemical compounds and protein is captured as a [CLS] vector of the top layer of each transformer, and these vectors are concatenated for further processing to predict DTIs. Arbitrary pretrained SMILES and protein transformers can be plugged into our framework. In this study, we used ChemBERTa and ProtBert to encode chemical compounds and amino acid sequences, respectively.

**Pretrained transformer for drugs.** To encode drug information in SMILES format, we used ChemBERTa, which is a pretrained RoBERTa model using PubChem [19] subsets, except for the 10 M subset of [8]. The maximum sequence length was 512 tokens with subword-level tokenization. The vocabulary size was 52 K.

**Pretrained transformer for proteins.** Protein information was encoded using ProtBert, which is a pretrained BERT model using datasets from UniRef [20] and BFD [21] containing up to 393 billion amino acids. In this model, the sequence length varies from 512 to 2000 with character-level tokenization. However, for computational efficiency, we set the maximum sequence length to 545, which covered 95% of the amino acid sequence length distribution. With our settings, an input amino acid sequence longer than 545 will be truncated. The vocabulary size was 30.

**Interaction layer.** The [CLS] obtained from the output of the last hidden layer is a vector representing the features of the sequence input to the BERT model. In the natural language model, this is used as a token for sequence classification. In our model, it was used to obtain features for a drug and features for a protein. To calculate the interaction prediction probability from the two [CLS] tokens containing this feature information, we first concatenated the two [CLS] tokens to create a single vector, *v*. Then, a Fully Connected Layer (FCN) was used to calculate the probability of drug and protein binding from the concatenated vector, *v*. FCN uses a vector as input and performs a linear operation to reduce the vector’s dimensions. The vector value of the dimension reduced through three FCNs is finally expressed as output, $v_{out}$, between 0 and 1 through the activation functions Relu and Tanh. After that, the interaction is indicated by 0 and 1 based on the threshold.
$$v_{1}=dropout\left(relu\left(FCN_{1}\left(v\right)\right)\right)$$
$$v_{2}=dropout\left(tanh\left(FCN_{2}\left(v_{1}\right)\right)\right)$$
$$v_{out}=FCN_{3}\left(v_{2}\right)$$

**Loss function.** The mean squared error (MSE) was used for the regression loss function. More specifically, the SmoothL1Loss function [22] was used, which has a similar calculation to MSE but is less sensitive to outliers.

### 2.3. Implementation Settings

**Dataset Settings.**Table 2 summarizes the training, validation, and test data from BIOSNAP, DAVIS, and BindingDB. The training and test data proportions followed the settings in [23]. The same data were used for accurate comparison with that paper; thus, the model was trained without processing the data further. In addition to MolTrans settings, in this study, we conducted novel experiments with an integration dataset constructed as described in Figure 2. In our integration dataset settings, all the training and validation DTI pairs from BIOSNAP, DAVIS, and BindingDB were put in one basket, and the pairs were fed to the training pipeline as well as being used for validation. However, individual DTI pairs from each dataset were tested in actual experiments.

**Model Parameters.** Our models were implemented with PyTorch. ChemBERTa and ProtBert in huggingface [7] were used for SMILES and amino acid sequence encoding, respectively. ChemBERTa used six layers and ProtBert used only the lower 18 layers. The maximum length of the sequence input to the encoder was 510 for drugs and 545 for proteins, accounting for 95% of the entire dataset. The dropout rate of the interaction layer was 0.1, and the batch size was 32. The optimizer used Adam, and the learning rate was set to $5\times 10^{-6}$.

**Hardware.** A single server with an AMD EPYC 7742 64-Core Processor CPU, 1024 GB RAM, and 4 × NVIDIA A100 Tensor Core GPUs was used for training and testing.

### 2.4. Model Performance Evaluation

A performance evaluation was carried out by comparing the proposed model with MolTrans, a DTI-prediction modality using a transformer-based model. The MolTrans dataset was employed to ensure accurate comparison. To confirm the effect of data augmentation on model performance, the training and validation data of three datasets were integrated and used for training. To check the effect of pretraining on performance, the proposed model was divided into four different submodels using the pretrained weights assigned to ChemBERTa and ProtBert. The models were trained on different datasets and the performances were compared; the optimally trained model is indicated in Table 3.

The evaluation metrics were the receiver operating characteristic area under the curve (*ROC-AUC*) and precision-recall-area under the curve (*PR-AUC*), and *sensitivity* was used as the main evaluation index. The predictive result is a continuous output expressed in binary form (using a threshold) that is then applied to the metric. To screen for candidate drug–target pairs in the DTI experiment, even if there were some pairs of negative interactions, a higher proportion of positive interaction pairs was better, so *sensitivity* is an important metric.
$$sensitivity=\frac{true\;positive}{true\;positive+false\;positive}$$

Table 4 provides the statistics. ROC indicates *sensitivity* to 1 specificity. A model with a high *ROC-AUC* will be less likely to rule out pairs with DTI when the threshold for determining whether pairs have DTI is high. The higher the *ROC-AUC*, the more likely it is to not rule out candidates for pairs with DTI. The PR-Curve is a graph showing the recall precision (*sensitivity*). Recall refers to the percentage of true positives. A higher *PR-AUC* means that there are fewer pairs without DTI among those that were initially judged to have DTI when the threshold used to classify them as having DTI is low. In actual experiments with pairs that are likely to have DTI, the higher the *PR-AUC*, the less probable it is that a pair without DTI will appear. Model performance was compared by evaluating the predictions using the Concordance Index(*CI*) [24] and $r_{m}^{2}$ coefficient [25] employed to evaluate the performance of the continuous output value.
$$CI=\frac{1}{Z}\sum_{\delta_{i}>\delta_{j}}^{}b\left(b_{i}-b_{j}\right)$$

$b_{i}$ and $b_{j}$ are the predictive values for a large affinity $\delta_{i}$ and a small affinity $\delta_{j}$, respectively, and $\delta_{i}$ > $\delta_{j}$ represents the case where the label of the *j*-th dataset sample is smaller than that of the *i*-th sample. *j* is an index from 0 to *i* − 1. *Z* is the normalization constant, and b(x) is expressed as a step function.
$$b\left(x\right)=\left\{\begin{array}{cc} 1 & \;\mathrm{if}\;x>0\\ 0.5 & \;\mathrm{if}\;x=0\\ 0 & \;\mathrm{if}\;x<0 \end{array}\right.$$

$r_{m}^{2}$ is a metric used for the validation of regression-based quantitative structure–activity relationship (QSAR) models, proposed by [25]
$$r_{m}^{2}=r^{2}\times \left(1-\sqrt{r^{2}-r_{0}^{2}}\right)$$

### 2.5. Additional Experiments: Prediction Dissociation Constant

Additional experiments were conducted to visualize the performance of the model. A drug (D) and a target (T) combine to form a drug–target complex (DT). Drugs and targets are partially bound in vivo. The dissociation constant ($K_{d}$) is the product of the D and T concentrations, defined using the following chemical equilibrium, and the $K_{d}$. $pK_{d}$ is the negative logarithm of $K_{d}$. The higher the $pK_{d}$, the higher the drug–target affinity.
D+T=DT
$$K_{d}=\frac{\left[D][T\right]}{\left[DT\right]}$$
$$pK_{d}=-logK_{d}$$

The model constructed above was modified to directly predict the $pK_{d}$ value; this yielded the dissociation constant. DAVIS and BindingDB datasets were used. A modified MolTrans (with the last sigmoid function removed) and a modified version of our model (with the last function, tanh, removed) were trained on both datasets. The predictive values were graphed.

### 2.6. Model Evaluation Using an External Dataset

To evaluate the model using an external dataset, 13 drug–protein pairs for which the protein/substrate relationships are well-known were selected based on the FDA guidelines on in vitro drug interactions [26]. No report on the interactions between drugs and targets has appeared; we thus explored drug–target interactions using public data, and we employed 13 drug–protein pairs featuring CYP protein and the drugs used by the FDA to evaluate drug interactions. The drugs and proteins are listed in Table 5. For model evaluation, the binding probabilities were predicted for each model (the MolTrans was trained with BindingDB and our models with BindingDB and the integrated dataset that featured pretrained weights) for all 13 drug–protein pairs. The DAVIS and BIOSNAP MolTrans datasets were excluded given their heterogeneity.

## 3. Results

### 3.1. Performance Evaluation

**DTI prediction performance.** To determine the effectiveness of the proposed models, we measured the *ROC-AUC*, *PR-AUC*, *sensitivity*, and *specificity* for models with the datasets described above. We compared the proposed method with another deep-learning DTI prediction model [23]. Table 4 shows the effectiveness of the proposed model and the baseline model.

The *ROC-AUC* and *sensitivity* scores of our models were high with all types of datasets. With the DAVIS dataset, there was a performance improvement of about 10% in *sensitivity*. In terms of the *PR-AUC*, the best performance was observed with the DAVIS and BindingDB datasets, and a comparable performance was achieved using the BIOSNAP dataset and MolTrans.

For *CI* and $r_{m}^{2}$, the proposed model achieved the highest performance with all datasets. Unlike the binary output metric of BindingDB, the model trained with the BindingDB dataset afforded the highest performance. This appears to be because the model used was based on *ROC-AUC*, and it is likely that the results are the same as those of the binary output metrics that can be obtained after hyperparameter tuning. After consideration of all the metrics used for performance comparisons, the FP-Model performed the best in most settings.

The DTI prediction performance using the integrated dataset was superior, particularly with DAVIS and BindingDB data. In these experiments, we simply put all training examples from the three different datasets into a single bucket and performed test predictions with each different dataset. Although the data imbalance could not be completely resolved, model performance was improved by increasing the amount of data that were used for training. With the existing statistical method, balanced data are essential; however, in deep learning models, performance can be improved by data augmentation.

### 3.2. Additional Experimental Visualization of the Prediction Dissociation Constant

To further evaluate the performance of the proposed model (as shown in Figure 3), we compared the predicted and actual binding affinity values for the DAVIS and BindingDB datasets. The models used for comparison were the MolTrans and FP-Model trained on each dataset. MolTrans divided the distribution of the predictions into two sets; the BindingDB dataset yielded prediction outputs that exceeded the actual values. The FP-Model distribution was linear by p=m within the range of actual values, indicating that prediction was better than that of MolTrans.

**Effects of pretraining.** To verify the effectiveness of pretraining, two different experiments were conducted to examine the learning time effect and data size effect. First, we compared the learning time and *ROC-AUC* across four different models (NP-Model-BDB, CP-Model-BDB, PP-Model-BDB, and FP-Model-BDB) with a fixed dataset (BindingDB dataset).
**Learning Time.**Figure 4a shows that FP-Model-BDB converged much faster and performed better than NP-Model-BDB which did not undergo any pretraining. The NP-Model-BDB took nine epochs (1170 s) to reach a *ROC-AUC* of 0.8326, whereas the FP-Model-BDB achieved the same performance after only one epoch. The figure indicates that chemical compound encoding pretraining affects the initial training stage, but protein encoding pretraining had more powerful effects in determining model performance at the final stage.**Data-Size.**Figure 4b shows that FP-Model-BDB achieved the same performance with fewer data than the NP-Model-BDB. FP-Model-BDB only needed 60% of the total data, whereas the NP-Model-BDB required 80% of the total training data.


**Figure 4 pharmaceutics-14-01710-f004:**
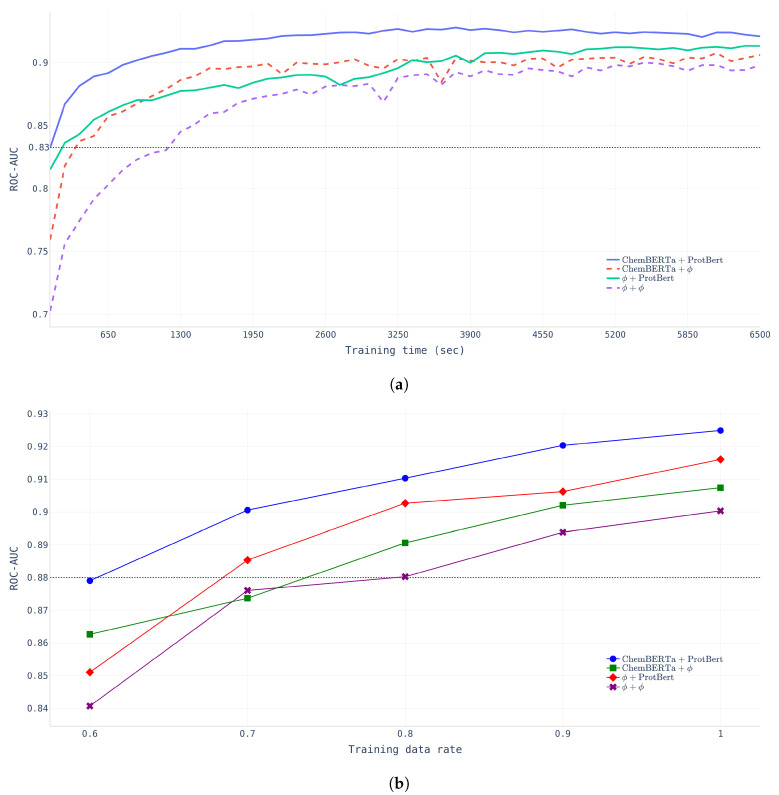
Pretraining effects with BindingDB data with four different models. (**a**) Learning time effect of pretraining. (**b**) Data-size effects of pretraining.

**Attention Visualization.** Another advantage of using a transformer model is that the visualization of internal causal behavior is more feasible than it is with other black-box-like neural network patterns. In this visualization process, attention weights, which play a role in determining the internal causal behavior of the transformer layer, are used. As the name suggests, the attention weight plays a role in determining how much the token value of the previous layer is reflected in the next layer. To visualize these attention weights, attention rollout [27] was used, and, as shown in Figure 5, an input token that had a high influence on the prediction of the model was identified.

Attention rollout is calculated by recursively performing matrix multiplication on the attention weights of layers in the transformer-based model. Through this process, when token embedding information is transmitted from the input layer to the upper layer, the bias of the weight applied to each token increases. By expressing this biased weight as an attention score for input tokens, you can intuitively check the token that had the highest influence on the output calculation.

Based on the visualized attention weights in Figure 5, it is possible to determine which input token has the greatest influence on prediction decisions with SMILES and amino acid sequence inputs into the model. From a biochemical viewpoint, drugs are extremely interactive with proteins at highly electronegative atoms, such as nitrogen (N) or sulfur (S), and the amino acids that interact with drugs also contain highly electronegative atoms. Figure 5 shows that the attention score learned by the model was consistent with human intuition. Knowing which tokens had a greater impact on DTI can help to speed up the drug discovery phase and researchers could find out which functional groups need to be improved.

**Model evaluation using an External dataset.** The model evaluations obtained using an external dataset are shown in Table 5. The MolTrans model trained on the BindingDB dataset yielded an average binding probability of 0.479 ± 0.204, and our models average binding probabilities are 0.687 ± 0.118 (NP-Model-BDB) and 0.730 ± 0.040 (FP-Model-BDB). NP-Model-INT and FP-Model-INT afforded average binding probabilities of 0.731 ± 0.025 and 0.731 ± 0.032, respectively.

## 4. Discussion

In a study of deep learning models that are useful in biochemistry, Alphafold and Evoformer [28] were used to determine the protein patterns produced by evolution. Evoformer trains to predict the protein structure by receiving input, such as similar amino acid sequences that appear in the process of evolution and the sequences that are to be predicted. From the above case, it can be inferred that there is an amino acid sequence pattern that exists in nature, and it can be learned through self-supervised learning using a transformer. In drug discovery, a molecule that binds to a protein is found by modifying the molecule through a lead optimization process. This is a pattern recognition process that has been performed by humans in the past, and ChemBERTa attempts to do this using a transformer model.

As shown in Figure 4, protein pretraining yielded the best performance. Proteins that exist in nature exhibit a specific pattern because they have survived the process of evolution, and it is assumed that this result was due to the influence of ProtBert, which had learned this pattern.

We confirmed that the type of dataset used to train the model affected the prediction performance. The BIOSNAP dataset had the greatest number of proteins among the three. In addition, the numbers of proteins and drugs were the most similar in this dataset. Therefore, the best *PR-AUC* results were obtained with this dataset. The DAVIS dataset is biased toward DTI values for tyrosinase (a protein involved in intracellular signaling) and its inhibitors. As a result, the poorest *PR-AUC* results were obtained with this dataset. There was a high likelihood of overfitting because only the binding probability for a specific protein and drug was learned. BindingDB contained the greatest number of interaction pairs among the three datasets. However, compared with DAVIS, the number of proteins was insufficient. Therefore, the *PR-AUC* values obtained with this dataset were lower than those obtained with the DAVIS dataset.

In this experiment, as for the model trained with the DAVIS dataset, two methods were used to prevent overfitting (given the small number of datasets). We confirmed that model performance improved on data augmentation; we integrated the pretrained models. In particular, the pretrained model was trained with large amounts of the SMILES and amino acid sequence datasets; it was thus possible to supplement training (which might have been inadequate using the drug–target pairs dataset alone). We employed the Masked Language Model (MLM) for pretraining. This pretraining differed from DTI prediction. When a model that has undergone pretraining engages in transfer learning, training is faster (fewer epochs) than that of a model trained without pretraining employing the same data; we confirmed that the final performance of the pretrained model was higher than that of the model lacking pretraining (Figure 4a). Thus, a language model pretrained with a rich dataset can increase the performance of a model trained using a small dataset. One of the most important effects of pretraining is generalization. An overfitted model exhibits high performance using a specific dataset but a poor performance on other datasets. After pretraining, we confirmed that our model performed better on datasets other than the reference dataset. Given the generalization effect, our model performed better on BindingDB (which contains the most diverse data) than other models. As a model affording good generalization is more robust to external data, this model more accurately selects the priority of future DTI experiments.

We compared the binding probabilities (using an external validation dataset) between well-understood drug–protein pairs. We found that pretraining and learned dataset expansion improved drug–target interaction predictions and could also find applications in other fields, such as natural language processes. Although the results are not shown, the MolTrans outputs varied by the learned dataset; this was not true of our model (which employed integrated data). Despite the promising results acquired with the external dataset, we used only 13 drug/protein pairs; it is difficult to acquire data that might validate the DB. More external verification data are required.

In terms of the visualization afforded by the additional experiment (Prediction Dissociation Constant), further training of the pretrained model rendered the predictions more continuous, confirming the effect of the above-mentioned pretraining. Our model was pretrained using a large dataset that lacked $pK_{d}$ information. After pretraining, knowledge of SMILES and amino acid sequences is included in the model; we presume that this previously learned knowledge enabled the continuous prediction of even $pK_{d}$.

## 5. Conclusions

When identifying screening hits during drug discovery, this model can determine the drugs that should be prioritized for drug–target interaction experiments. During the lead optimization of a drug, it is possible to predict affinity before synthesizing the drug, although high affinity does not always indicate that a drug is effective. This method can be used to find drugs with similar structures but slightly different affinities. Learning the patterns of various proteins is thought to be important in the screening process for drug discovery. It was also confirmed that the *PR-AUC* would improve as the number of drugs approached the number of proteins. The *PR-AUC* plays an important role in filtering out the drug–target pairs that will fail during screening. Therefore, this is an important metric for reducing cost when performing large-scale DTI experiments. 

## Figures and Tables

**Figure 1 pharmaceutics-14-01710-f001:**
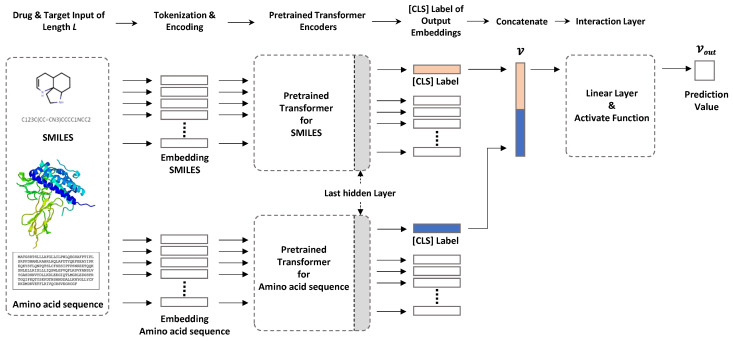
Our model configuration. The sequence information of compounds and proteins is captured as a [CLS] vector of the last hidden layer in each pretrained transformer. Each piece of captured sequence information is concatenated and input to the interaction layer, and the DTI prediction value is output.

**Figure 2 pharmaceutics-14-01710-f002:**
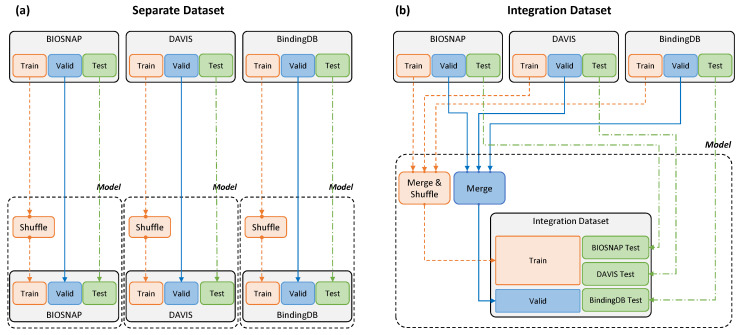
Separate and integration datasets. (**a**) A separate dataset was input by dividing the three datasets into training and validation test data for each model. (**b**) The integration dataset was used for model training by merging the training and validation data from the three datasets, and evaluation was conducted in the same way as for the separate dataset.

**Figure 3 pharmaceutics-14-01710-f003:**
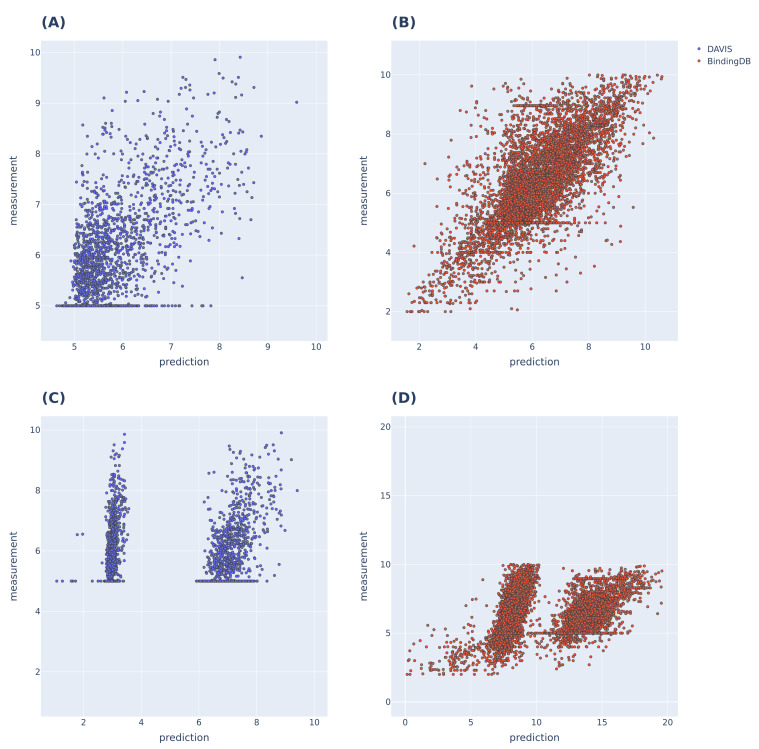
$pK_{d}$ prediction results of MolTrans and FP-Models trained with DAVIS and BindingDB, respectively. (**A**,**B**) are the $pK_{d}$ predictions of the FP-Model; linearity is evident within the label range. (**C**,**D**) are the $pK_{d}$ predictions of MolTrans, and the prediction distributions were divided into two sets for both datasets; the BindingDB dataset predicted a higher $pK_{d}$ value than the maximum value of the label range.

**Figure 5 pharmaceutics-14-01710-f005:**
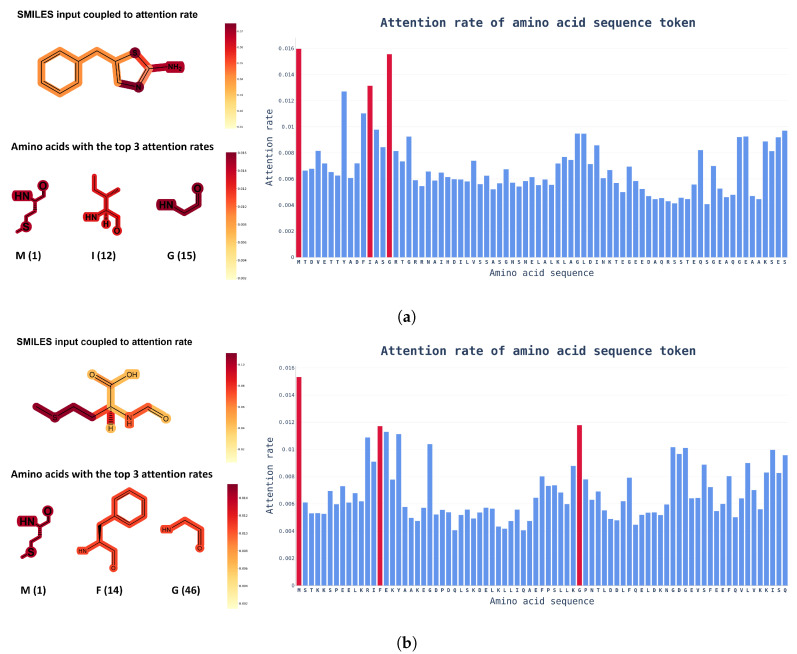
Examples of chemical structural formulas of drugs and proteins with attention weights in the integration dataset. The color bar on the right indicates the attention scores of ChemBERTa and ProtBert. In the amino acid sequence, only the chemical structural formula of the top three amino acids with the highest attention weights are shown, along with position information for readability. (**a**) Attention score of a protein paired with 5-benzylthiazol-2-amine (DTI prediction score: 0.7744); (**b**) Attention score of a protein paired with formyl-L-methionine (DTI prediction score: 0.7126).

**Table 1 pharmaceutics-14-01710-t001:** Dataset statistics.

Dataset	Drug	Proteins	Interactions
BIOSNAP	4510	2181	27,482 (13,741/13,741)
DAVIS	68	379	11,103 (1506/9597)
BindingDB	10,665	1413	32,601 (9166/23,435)

*Note*: Values in parentheses indicate the number of positive and negative interactions.

**Table 2 pharmaceutics-14-01710-t002:** Training, validation, and test data size for each dataset.

Dataset	Training	Validation	Test
BIOSNAP	19,238	2748	5496
DAVIS	2086	3006	6011
BindingDB	12,668	6644	13,289
Integration	33,992	12,398	(5496/6011/13,289)

**Table 3 pharmaceutics-14-01710-t003:** Model utilities by the configurations and the training datasets used.

Dataset	BIOSNAP (BS)	DAVIS (DV)	BindingDB (BDB)	Integration (INT)
Non-pretrained (NP) (ϕ + ϕ)	NP-Model-BS	NP-Model-DV	NP-Model-BDB	NP-Model-INT
ChemBERTa-pretrained (CP) (ChemBERTa + ϕ)	CP-Model-BS	CP-Model-DV	CP-Model-BDB	CP-Model-INT
ProtBERT-pretrained (PP) (ϕ + ProtBERT)	PP-Model-BS	PP-Model-DV	PP-Model-BDB	PP-Model-INT
Full-pretrained (FP) (ChemBERTa + ProtBERT)	FP-Model-BS	FP-Model-DV	FP-Model-BDB	FP-Model-INT

*Note*: *ϕ* indicates the model that did not employ the pretrained weights of the ChemBERTa and ProtBert models.
Integration refers to a model trained by merging the three datasets.

**Table 4 pharmaceutics-14-01710-t004:** Performance comparison. *ROC-AUC*, *PR-AUC*, *Sensitivity*, and *Specificity* were measured over five random runs to accurately compare the performance with that of the cited MolTrans paper. *CI* and $r_{m}^{2}$ compare the performance of the proposed model with the performance when a metric is added to the published MolTrans model. The highest performances among the metrics used to evaluate each dataset are shown in bold on a gray background.

Method	ROC-AUC	PR-AUC	Sensitivity	Specificity	CI	$r_{m}^{2}$
Dataset 1. BIOSNAP						
MolTrans	0.895 ± 0.002	**0.901 ± 0.004**	0.775 ± 0.032	0.851 ± 0.014	0.889	0.449
NP-Model-BS	0.882 ± 0.004	0.871 ± 0.015	0.779 ± 0.020	0.850 ± 0.012	0.895	0.428
CP-Model-BS	0.881 ± 0.009	0.859 ± 0.017	0.811 ± 0.018	0.835 ± 0.008	0.891	0.406
PP-Model-BS	0.893 ± 0.003	0.874 ± 0.006	0.803 ± 0.033	0.851 ± 0.019	0.896	0.425
FP-Model-BS	**0.914 ± 0.006**	0.900 ± 0.007	**0.862 ± 0.025**	0.847 ± 0.007	**0.913**	**0.467**
NP-Model-INT	0.877 ± 0.007	0.860 ± 0.010	0.785 ± 0.007	0.842 ± 0.008	0.897	0.421
CP-Model-INT	0.875 ± 0.006	0.851 ± 0.009	0.775 ± 0.023	0.844 ± 0.016	0.885	0.401
PP-Model-INT	0.895 ± 0.003	0.880 ± 0.008	0.802 ± 0.018	0.852 ± 0.009	0.896	0.435
FP-Model-INT	0.910 ± 0.012	0.897 ± 0.014	0.830 ± 0.029	**0.863 ± 0.011**	0.911	0.447
Dataset 2. DAVIS						
MolTrans	0.907 ± 0.002	0.404 ± 0.016	0.800 ± 0.022	0.876 ± 0.013	0.903	0.156
NP-Model-DV	0.870 ± 0.003	0.283 ± 0.005	0.738 ± 0.030	0.871 ± 0.026	0.875	0.118
CP-Model-DV	0.882 ± 0.006	0.250 ± 0.023	0.744 ± 0.021	0.888 ± 0.019	0.878	0.117
PP-Model-DV	0.866 ± 0.003	0.263 ± 0.007	0.747 ± 0.020	0.856 ± 0.012	0.864	0.115
FP-Model-DV	0.920 ± 0.002	0.395 ± 0.007	0.824 ± 0.026	**0.889 ± 0.015**	0.917	0.167
NP-Model-INT	0.899 ± 0.008	0.322 ± 0.030	0.814 ± 0.039	0.857 ± 0.028	0.892	0.141
CP-Model-INT	0.904 ± 0.011	0.351 ± 0.035	0.814 ± 0.030	0.859 ± 0.020	0.917	0.169
PP-Model-INT	0.923 ± 0.005	0.417 ± 0.028	0.844 ± 0.017	0.876 ± 0.021	0.916	0.162
FP-Model-INT	**0.942 ± 0.005**	**0.517 ± 0.017**	**0.903 ± 0.017**	0.866 ± 0.015	**0.940**	**0.201**
Dataset 3. BindingDB						
MolTrans	0.914 ± 0.001	0.622 ± 0.007	0.797 ± 0.005	0.896 ± 0.007	0.899	0.267
NP-Model-BDB	0.891 ± 0.005	0.515 ± 0.014	0.774 ± 0.012	0.897 ± 0.013	0.899	0.309
CP-Model-BDB	0.914 ± 0.003	0.585 ± 0.021	0.803 ± 0.011	0.904 ± 0.010	0.907	0.320
PP-Model-BDB	0.897 ± 0.003	0.557 ± 0.013	0.775 ± 0.019	0.900 ± 0.009	0.913	0.324
FP-Model-BDB	0.922 ± 0.001	0.623 ± 0.010	**0.814 ± 0.025**	0.916 ± 0.016	**0.927**	**0.365**
NP-Model-INT	0.904 ± 0.001	0.574 ± 0.008	0.766 ± 0.015	0.910 ± 0.015	0.907	0.315
CP-Model-INT	0.909 ± 0.005	0.600 ± 0.019	0.787 ± 0.008	0.907 ± 0.008	0.918	0.330
PP-Model-INT	0.918 ± 0.001	0.607 ± 0.012	0.787 ± 0.014	0.920 ± 0.010	0.916	0.344
FP-Model-INT	**0.926 ± 0.001**	**0.639 ± 0.018**	0.802 ± 0.022	**0.928 ± 0.013**	0.926	0.362

**Table 5 pharmaceutics-14-01710-t005:** Prediction of 13 well-known drug/protein interactions.

CYP Subtype (Targets)	Drugs	MolTrans	NP-Model-BDB	FP-Model-BDB	NP-Model-INT	FP-Model-INT
1A2	Phenacetin	0.391	0.745	0.774	0.767	0.764
1A2	7-Ethoxyresorufin	0.255	0.747	0.786	0.762	0.784
2B6	Efavirenz	0.755	0.728	0.755	0.687	0.749
2B6	Bupropion	0.524	0.727	0.757	0.712	0.733
3A4	Midazolam	0.352	0.707	0.694	0.719	0.721
3A4	Testosterone	0.125	0.741	0.699	0.714	0.662
2C8	Paclitaxel	0.823	0.698	0.676	0.722	0.732
2C8	Amodiaquine	0.631	0.694	0.712	0.748	0.759
2C9	S-Warfarin	0.546	0.725	0.655	0.701	0.705
2C9	Diclofenac	0.556	0.645	0.758	0.726	0.711
2C19	S-Mephenytoin	0.313	0.738	0.742	0.740	0.744
2D6	Bufuralol	0.475	0.355	0.749	0.756	0.699
2D6	Dextromethorphan	-	0.490	0.740	0.750	0.740
	**Average BP**	0.479	0.687	0.730	0.731	0.731

## Data Availability

The original contributions presented in the study are included in the article and available on https://github.com/hskang0906/DTI-Prediction.git (accessed on 22 July 2022).

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
