# Peer review of "Fine-tuning of BERT Model to Accurately Predict Drug–Target Interactions"

_pharmaceutics, 2022, doi:10.3390/pharmaceutics14081710_

Round 1

Reviewer 1 Report

This is a really interesting and important report of work on using pretrained BERT models to accelerate DTI training and obtain good prediction performance. The organization of the article is very clear and convincing except for some space of improvements on method description. Overall, it will be an impactful piece of work inspiring researchers in the field to adopt this methodology.

The English writing of this article, however, requires significant improvement and actually it might negatively affect the audience when they’re reading it, so a thorough review and maybe professional service is needed to make this article easier to understand.

Comments

  1. Title: It’s up to the authors but would that be better to just call it “The use of pretrained Bidirectional Encoder Representations from Transformers (BERT) model to accurately predict drug-target interactions”?
  2. The article needs a very thorough review in terms of English writing. A few examples:
  1. Line 1: “Identification of optimal drug candidates is very important drug discovery and Researchers …”: Should be “in drug discovery” and should separate into two sentences.
  2. Line 3: “Given the emerging utilities of models”: Should be “emerging applications of models”?
  3. Line 4: NLP stands for Natural language processing? Also it is not clear what it means by “models pretrained using NLPs” — Should that be pretrained NLP models? The following sentence “such models of … are under development” — Not clear what this sentence means.
  4. There are just too many sentences which are hard to understand. I would recommend the authors to get support from professional English reviewing services to make the article natural and easy to understand for audience around the world.
  1. 9 citations: [4], [5], [7], [8], [15], [16], [17], [26], [27] out of 28, are all arXiv preprints. I would double check if these have peer-reviewed versions even if they’re accepted conference abstracts. Having so many citations that are not peer-reviewed worries me a little about the soundness of these references.
  2. Line 28: “The inputs are drug and protein data”: Could the authors elaborate this? What protein data? Also, are all drug targets proteins?
  3. Line 32: Is there any literature supporting that pretrained models are “optimal”?
  4. Line 42: It should be “the dataset used to train the ML model”?
  5. Line 45: Is it possible to introduce what are BIOSNAP, DAVIS, and Binding DB?
  6. Line 99: Is it possible for the authors to briefly introduce what is CLS?
  7. Figure 1 title: It is okay to just use “Our model configuration” as the figure title, but it might be better to highlight the special or distinct feature of this architecture. Many readers might jump straight to this figure right after they open this article, and having a descriptive title will be helpful for those readers.
  8. Line 127-130: I believe this is one of the innovation of this network in the article and this paragraph deserves more description and explanation. Could the authors explain why this 3 layer network is chosen? Also “The vector is processed as a fully connected layer (FCN)” — Isn’t it 3 layers based on the equations or I misunderstood the structure? The author might also want to mention briefly how these layers are trained here.
    1. I noticed section 3.1 is actually method but in the Results section. It might be better to move it to the 2. Datasets and Methods section so that the readers could learn how the training process works.
  9. Line 166: Star sign in equation: Either remove it (or use multiplication sign) if it means multiplication, or explain what star sign means.
  10. Line 255 and Figure 5: This is very interesting and important insight from the model and gives interpretability to the authors model. Therefore I believe it deserves more description and highlight. For example, is it possible to explain better what is attention rollout? And maybe compare with other attention metrics used in deep neural networks and why choosing the current metric? Figure 5 shows two examples of attention score of pairs — Is there any summary statistics or overall pattern showing M, I, F, G amino acids are important within all the proteins in the dataset?
  11. Is there any possibility for the authors to share or point audience to the source code (e.g. GitHub) so that interested readers could reproduce the authors’ work?

Reviewer 2 Report

In this paper the authors worked on development of drug-target interactions pretrained with ChemBERTa and ProtBERT that analyses information on chemical compounds and proteins respectively confirming that the final performance of the pretrained model was higher than that of the model lacking pretraining. Also the type of dataset used to train the model showed to affect the prediction, a language model pretrained with a rich dataset can increase the performance of a model trained using a small dataset.

Authors provided convenient attention visualization that highly contributes to the quality of this work, however as authors mentioned much more external verification data are required, and I hope the authors will continue developing model validation procedures in their future work.

Author Response

Thank you for your comments. we will proceed with the reviewer's opinion in the future revision.

This manuscript is a resubmission of an earlier submission. The following is a list of the peer review reports and author responses from that submission.

Round 1

Reviewer 1 Report

This Report consists of an application of (a combination of)  pre-given programs.

The authors could not demonstrate any significant progress.

In order for a reasonable review the authors should:

Demonstrate that they have understood BERT in detail by describing it in a scientific way.

Describe Transformers in a scientific way.

Examine statiscically distributions, distances et. the statistics of the intermediate results, inputs and outputs.

Marking the maxima in Table 3 is ridiculous. There are scientific methods to prove or in this case disprove that  a 91.4 % performance  (their algorithm) is no better than the others with all algorithms within a range of 85...91 %

Are you aware of the mathematical properties of "linear layers"? what use is the plural?

Reviewer 2 Report

An interesting study worth to be published.

Author Response

we would like to appreciate your comments.

Reviewer 3 Report

The paper "Accurate and precise drug-target interaction prediction using pretrained transformer encoders" is devoted to a topical and important problem. The authors, using Bidirectional Encoder Representations from Transformers (BERT) with ChemBERTa model and a ProtBert model, were able to build quite adequate classification models for predicting drug-target interaction. A positive feature of the work is the convenient and visualized interpretability of the constructed models demonstrated by the authors.

There are no fundamental comments on the manuscript of the paper; however, the authors should clarify some details of the work:

1. The DAVIS and BindingDB datasets are significantly unbalanced - there are significantly more negative interactions than positive ones (Table 1). It is necessary to understand whether the authors used special procedures to balance the data. Perhaps the best classification results for the BIOSNAP dataset are due to the fact that it is fully balanced.

2. The article does not contain any information regarding the applicability domain of the models used

Thus, the manuscript can be published in Pharmaceutics after taking into account these comments.
